**Data Availability Statement:** Data cannot be shared publicly because the informed consent that was signed by the study participants is not

# The relation between urinary sodium and potassium excretion and risk of cardiovascular events and mortality in patients with cardiovascular disease

Eline H. Groenland[1], Jean-Paul Vendeville[1], Michiel L. Bots[2], Gert Jan de Borst[3], Hendrik M. Nathoe[4], Ynte M. Ruigrok[5], Peter J. Blankestijn[6], Frank L. J. Visseren[1], Wilko Spiering[1]*, on behalf of the UCC-SMART Study Group[¶]

1 Department of Vascular Medicine, University Medical Center Utrecht, Utrecht University, Utrecht, The Netherlands, 2 Julius Center for Health Sciences and Primary Care, University Medical Center Utrecht, Utrecht University, Utrecht, The Netherlands, 3 Department of Vascular Surgery, University Medical Centre Utrecht, Utrecht, The Netherlands, 4 Department of Cardiology, University Medical Center Utrecht, Utrecht, The Netherlands, 5 Department of Neurology, University Medical Center Utrecht, Utrecht, The Netherlands, 6 Department of Nephrology, University Medical Center Utrecht, Utrecht University, Utrecht, The Netherlands

¶ Membership of the UCC-SMART Study Group is listed in the Acknowledgments.

* W.Spiering@umcutrecht.nl

## Abstract

### Background

Most evidence on the relationship between sodium and potassium intake and cardiovascular disease originated from general population studies. This study aimed to evaluate the relation between estimated 24-hour sodium and potassium urinary excretion and the risk of recurrent vascular events and mortality in patients with vascular disease.

### Methods

7561 patients with vascular disease enrolled in the UCC-SMART cohort (1996–2015) were included. Twenty-four hour sodium and potassium urinary excretion were estimated (Kawasaki formulae) from morning urine samples. Cox proportional hazard models with restricted cubic splines were used to evaluate the relation between estimated urinary salt excretion and major adverse cardiovascular events (MACE; including myocardial infarction, stroke, cardiovascular mortality) and all-cause mortality.

### Results

After a median follow-up of 7.4 years (interquartile range: 4.1–11.0), the relations between estimated 24-hour sodium urinary excretion and outcomes were J-shaped with nadirs of 4.59 gram/day for recurrent MACE and 4.97 gram/day for all-cause mortality. The relation between sodium-to-potassium excretion ratio and outcomes were also J-shaped with nadirs of 2.71 for recurrent MACE and 2.60 for all-cause mortality. Higher potassium urinary excretion was related to an increased risk of both recurrent MACE (HR 1.25 per gram potassium

compliant with publishing individual data in an open access institutional repository or as Supporting information files with the published paper. However, a data request can be sent to the SMART Steering Committee at uccdatarequest@umcutrecht.nl.

**Funding:** The author(s) received no specific funding for this work.

**Competing interests:** The authors have declared that no competing interests exist.

excretion per day; 95%CI 1.13–1.39) and all cause-mortality (HR 1.13 per gram potassium excretion per day; 95%CI 1.03–1.25).

## Conclusions

In patients with established vascular disease, lower and higher sodium intake were associated with higher risk of recurrent MACE and all-cause mortality. Higher estimated 24-hour potassium urinary excretion was associated with a higher risk of recurrent MACE and all-cause mortality.

## Introduction

Blood pressure (BP) control is an essential target for the prevention and management of recurrent cardiovascular disease (CVD) in patients with established vascular disease. In adults with and without hypertension, higher sodium intake is linearly associated with higher BP levels [1, 2], and therefore most treatment guidelines advocate dietary sodium restriction to levels between 1.5 and 2.4 g per day to lower the risk of (recurrent) CVD [3–5].

However, previous cohort studies evaluating the association between sodium intake and CV events in primary prevention populations have shown conflicting results. While some studies report a neutral or positive linear association between sodium intake and CVD and total mortality [6–8], others demonstrate a J- or U-shaped relationship between estimated sodium intake and CVD risk with lower and higher sodium intake both being associated with higher risk of CVD, all-cause mortality, and longevity [9–12]. Thus, guideline recommendations on dietary sodium intake conflict with findings from several observational studies regarding CVD risk.

In contrast to sodium, higher potassium intake has been inversely related to BP levels and may have a protective effect, thereby modifying the association between sodium intake, BP and CVD [10, 13]. Consequently, both the World Health Organization (WHO) and recent guidelines on the primary prevention of CVD recommend an intake of at least 3.5 grams per day [4, 5, 14]. In addition, emerging evidence suggest that the sodium-to-potassium excretion ratio represents a more important risk factor for CVD than sodium and potassium separately [6, 15]. Since most of the evidence on the relationship between sodium and potassium intake and CVD originated from general population studies, the question is whether the above guideline recommendations can be applied to patients with established vascular disease. Clarifying the optimal dietary sodium and potassium intake is especially important in patients with clinical manifest arterial disease who are most likely to receive recommendations regarding dietary salt intake.

Hence, the aim of this study was to examine the relation between estimates of 24-hour sodium and potassium urinary excretion (as proxies for dietary intake), as well as their ratio, and the risk of recurrent major adverse cardiovascular events (MACE) and all-cause mortality in a high-risk population cohort with stable CVD.

## Methods

### Study design and participants

Patients originated from the Utrecht Cardiovascular Cohort-Second Manifestation of ARTerial disease (UCC-SMART) cohort. The UCC-SMART cohort is an ongoing, prospective

cohort study starting from 1996 and comprised of 18 to 79-year-old patients referred to the University Medical Center Utrecht (UMCU), the Netherlands, for management of atherosclerotic disease or cardiovascular risk factors. A detailed description of the study rationale and design has been previously described [16]. The study is in accordance with the 1964 Helsinki declaration, was approved by the institutional review board of the Utrecht University Medical Center, and all patients gave written informed consent.

For the current study, patients with established vascular disease (coronary heart disease, cerebrovascular disease, peripheral arterial disease or abdominal aortic aneurysm) at baseline between January 1996 and February 2015 were included (n = 7561).

## Baseline assessment

At baseline, the patients underwent a standardized vascular screening protocol consisting of a health questionnaire including medical history and risk factors, physical examination and laboratory testing.

Office BP was measured with a nonrandom sphygmomanometer (Iso-Stabil 5; Speidel & Keller, Jungingen, Germany) three times simultaneously at the right and left upper arm in an upright position with an interval of 30 seconds. The mean of the last two BP measurements from the arm with the highest BP was used. Hypertension was defined as a prescription of antihypertensive medication and/or an office systolic BP of ≥140 or diastolic BP of ≥90 mmHg.

Laboratory blood testing was performed in fasting state for total cholesterol, triglycerides, high-density lipoprotein (HDL) cholesterol, creatinine, and high-sensitivity C-reactive protein (CRP). Low-density lipoprotein (LDL) cholesterol was calculated using the Friedewald formula [17] up to a plasma triglycerides level of 9 mmol/L [18]. The estimated glomerular filtration rate (eGFR) was calculated using the Chronic Kidney Disease Epidemiology Collaboration (CKD-EPI) formula [19].

Upon arrival at the study clinic, usually in the morning, a urine sample was collected in fasting state and stored at -20˚C. Urinary sodium and potassium levels were measured using an ARCHITECT *ci*8200 analyzer (Abbott Laboratories, Lake Bluff, Illinois, USA). The coefficient of variation for both sodium and potassium was 3%, and 6% for creatinine. The Kawasaki formula was used to estimate 24-hour sodium and potassium urinary excretion from a fasting morning urine sample, and these estimates were used as proxies for sodium and potassium intake [20] (S1 Table). We chose to use the Kawasaki formula to allow comparability between this and previous studies and because this formula is considered the least biased method for estimating 24-hour sodium excretion compared to other formula-based approaches [21].

## Outcome assessment

Patients received a bi-annual health questionnaire concerning hospitalizations and outpatient clinic visits. Outcomes of interest for this study were first occurrence of myocardial infarction, stroke, vascular death, and a composite of these events (all vascular events). All-cause mortality was recorded as well. Definitions of events are shown in S2 Table. When a possible event was reported, hospital records including radiology examinations, laboratory reports, and hospital discharge letters, were collected. Death and cause of death were reported by relatives of the participant, the general practitioner, or the vascular specialist. The medical records and information from the questionnaire and/or the family were subsequently assessed by three separate physicians from the study end-point committee. Duration of follow-up was defined as the time between study enrollment and first cardiovascular event or death from any cause, date of loss to follow-up (n = 407 (5.4%)), or the preselected date of March 1st, 2015.

## Statistical analysis

Baseline characteristics are presented stratified in quintiles of estimated 24-hour sodium and potassium urinary excretion. Because complete case analysis would lead to loss of statistical power and possibly bias [22], missing data of determinants and possible confounders (urine sodium (n = 510, 6.7%), potassium (n = 440, 5.8%), urine creatinine (n = 200, 2.6%), CRP (n = 179, 2.4%) and ≤1% for other variables) was imputed using single regression imputation (aregImpute-algorithm in R, Hmisc package).

Linear regression models were fitted to examine the association between estimated 24-hour sodium and potassium urinary excretion and blood pressure. Restricted cubic-spline functions with four knots were used to explore the shape of the association between baseline salt measures (estimated 24-hour sodium urinary excretion, estimated 24-hour potassium urinary excretion, and the ratio between the two) and the outcomes [23]. Based on visual inspection of the restricted-cubic spline plots, a quadratic relation between outcomes and estimated 24-hour sodium urinary excretion and the sodium-to-potassium excretion ratio seemed present. Hence, we fitted multivariable Cox proportional-hazards models, including linear and quadratic terms for estimated 24-hour sodium urinary excretion and the sodium-to-potassium excretion ratio. As the restricted cubic-spline plots of the relationship between the estimated 24-hour potassium urinary excretion and outcomes showed no sign of non-linearity, these Cox proportional-hazards model only included a linear term. Proportional hazards assumptions were tested by visual inspection of Schoenfeld residuals plots and no violation was observed.

Analyses were adjusted for age, sex, body mass index (BMI), smoking, presence of diabetes, eGFR, and non-HDL cholesterol. The p-values of the effects of baseline salt measures on the occurrence of vascular events and mortality were based on the $\chi^2$ statistic. Nadirs (value of salt measures associated with lowest hazard) were derived for the non-linear relations. Hazard ratios (HR's) with 95% confidence intervals (CIs) were reported for the linear associations. Nadirs were derived as the minimum of the quadratic function that models the relation between outcomes and baseline salt measures. For graphic representation of the relationship between estimated sodium urinary excretion and the sodium-to-potassium excretion ratio and cardiovascular events and mortality, hazard ratios and 95% CIs were plotted, taking the corresponding nadir as a reference.

We performed interaction analyses for key characteristics that might modify the association between salt measures and CV events (age (<65 years versus ≥ 65 years), sex, use of blood-pressure lowering medication, and hypertension). Moreover, we tested the interaction between estimated 24-hour sodium and potassium urinary excretion. When a significant interaction was found, the analyses were stratified according to the effect modifying characteristic.

Sensitivity analyses were performed to evaluate the likelihood of reverse causality. Because reverse causality, if present, affects short-term rather than long-term results, analyses were repeated excluding patients with events within 1, 2, and 5 year(s) after inclusion. Furthermore, we performed analyses excluding patients treated with loop diuretics at baseline since this is often prescribed in the treatment of heart failure and often also accompanied by sodium restriction. Lastly, to evaluate whether patients with low levels of salt excretion had lower survival rates in the first years of follow-up, Kaplan-Meier survival curves were plotted by quintile of each salt measure (estimated 24-hour sodium excretion, estimated potassium excretion, and stage-to-potassium ratio) for recurrent CVD and all-cause mortality.

All analyses were performed with R statistical software (Version 3.5.1; R foundation for Statistical Computing, Vienna, Austria). All p-values were two-tailed, with statistical significance set at 0.05.

## Results

### Baseline characteristics

Baseline characteristics for all subjects categorized by quintile of estimated 24-hour sodium urinary excretion and estimated 24-hour potassium urinary excretion are summarized in Table 1 and S3 Table, respectively. The mean estimated 24-hour sodium urinary excretion was 4.91 g/day (standard deviation (SD) 1.41), and the mean estimated 24-hour potassium urinary excretion was 2.18 g/day (SD 0.53). Patients with low estimated 24-hour sodium and potassium urinary excretion were younger, had lower BMI, were less likely to have a history of diabetes mellitus or coronary artery disease; and generally had a lower blood pressure. Furthermore, they were more likely to be current smokers, have a history of cerebrovascular disease, and use diuretics.

During a median follow-up of 7.4 years (interquartile range (IQR): 4.1–11.0 years; 58,386 person-years), the composite outcome of myocardial infarction, stroke, or vascular death occurred in 1332 patients. A total of 1502 deaths were reported.

### Relation between estimated 24-hour sodium and potassium excretion and blood pressure

Adjusted linear regression models assessing the relationship between baseline estimated 24-hour sodium urinary excretion and baseline blood pressure showed that every 1 g/day increase of sodium urinary excretion was associated with a higher mean (95% CI) systolic blood pressure and diastolic blood pressure of 1.28 mmHg (0.95–1.62) and 0.46 mmHg (0.28–0.65), respectively. Every 1 g/day increase of potassium urinary excretion was also associated with a higher mean (95% CI) systolic blood pressure and diastolic blood pressure of 1.04 mmHg (0.15–1.93) and 0.61 mmHg (0.11–1.11), respectively.

### Relation between 24-hour sodium excretion and recurrent cardiovascular events and all-cause mortality

The relationship between estimated 24-hour sodium urinary excretion and the incidence of vascular events followed a J-shaped curve, with increased hazard ratios at low and high sodium urinary excretions. This was initially explored by a Cox proportional-hazards model with restricted cubic splines (S1 Fig) and confirmed by a non-linear Cox proportional-hazards model including linear and quadratic sodium urinary excretion terms (p = 0.02; non-linear term p<0.01) (Fig 1A). Similarly, the relationship between estimated 24-hour sodium urinary excretion and all-cause mortality followed a J-shaped curve (Fig 1B; p<0.01; non-linear term p<0.01). The nadir for vascular events was 4.59 g/day and 4.97 g/day for all-cause mortality. No association was found between estimated 24-hour sodium urinary excretion and the occurrence of stroke (p = 0.91, non-linear term p = 0.61) (S2 Fig) and the occurrence of myocardial infarction (p = 0.97; non-linear term p = 0.76) (S3 Fig). Still, the relationship between sodium urinary excretion and vascular mortality was J-shaped (p<0.01, non-linear term p<0.01, nadir 4.98) (S4 Fig).

### Relation between 24-hour potassium excretion and recurrent cardiovascular events and all-cause mortality

No evidence of non-linearity in the relations between estimated 24-hour potassium urinary excretion and any outcome was found in the fully adjusted models; all non-linear p-values were >0.05 (S1 Fig). Therefore, Cox proportional-hazards models that investigated the relation between potassium urinary excretion and recurrent MACE and all-cause mortality only

**Table 1. Baseline characteristics of all participants, according to estimated 24-hour sodium excretion.**

| | Overall | Estimated urinary sodium excretion, g/day; quintiles | | | | |
|---|---|---|---|---|---|---|
| | | Q1 | Q2 | Q3 | Q4 | Q5 |
| Range quintiles (g/day) | | [1.28–3.73] | [3.74–4.47] | [4.48–5.13] | [5.14–5.97] | [5.98–16] |
| Mean Sodium (g/day) | 4.9 ± 1.4 | 3.1 ± 0.5 | 4.1 ± 0.2 | 4.8 ± 0.2 | 5.5 ± 0.2 | 7.0 ± 1.0 |
| | n = 7561 | n = 1513 | n = 1512 | n = 1512 | n = 1512 | n = 1512 |
| Male sex | 5574 (74%) | 864 (57%) | 1036 (69%) | 1153 (76%) | 1227 (81%) | 1294 (86%) |
| Age (years) | 60 ± 10 | 58 ± 11 | 60 ± 10 | 60 ± 10 | 61 ± 10 | 61 ± 10 |
| Current smoker | 2396 (32%) | 606 (40%) | 487 (32%) | 496 (33%) | 414 (27%) | 393 (26%) |
| *Physical examination* | | | | | | |
| Body mass index (kg/m2) | 26.8 ± 4.0 | 26.0 ± 4.1 | 26.3 ± 3.8 | 26.7 ± 3.7 | 27.2 ± 3.9 | 28.0 ± 4.3 |
| Systolic blood pressure (mmHg) | 140 ± 21 | 137 ± 20 | 139 ± 21 | 140 ± 20 | 141 ± 21 | 143 ± 21 |
| Diastolic blood pressure (mmHg) | 81 ± 11 | 80± 11 | 80 ± 11 | 81 ± 11 | 82 ± 11 | 82 ± 11 |
| *History of vascular disease* | | | | | | |
| Diabetes mellitus | 1327 (18%) | 218 (14%) | 221 (15%) | 225 (15%) | 287 (19%) | 376 (25%) |
| Coronary artery disease | 4576 (61%) | 784 (52%) | 880 (58%) | 930 (62%) | 990 (65%) | 992 (66%) |
| Peripheral artery disease | 1408 (19%) | 312 (21%) | 290 (19%) | 264 (17%) | 273 (18%) | 269 (18%) |
| Cerebrovascular disease | 2247 (30%) | 545 (36%) | 468 (31%) | 438 (29%) | 397 (26%) | 399 (26%) |
| Abdominal aortic aneurysm | 650 (9%) | 124 (8%) | 126 (8%) | 107 (7%) | 132 (9%) | 161 (11%) |
| *Laboratory values* | | | | | | |
| Potassium excretion (g/day) | 2.2 ± 0.5 | 1.9 ± 0.5 | 2.0 ± 0.4 | 2.1 ± 0.5 | 2.3 ± 0.5 | 2.6 ± 0.6 |
| Total cholesterol (mmol/L) | 4.9 ± 1.2 | 5.0 ± 1.2 | 4.9 ± 1.2 | 4.8 ± 1.2 | 4.8 ± 1.2 | 4.8 ± 1.2 |
| HDL-cholesterol (mmol/L) | 1.2 ± 0.4 | 1.3 ± 0.4 | 1.3 ± 0.4 | 1.2 ± 0.4 | 1.2 ± 0.3 | 1.2 ± 0.4 |
| LDL-cholesterol (mmol/L) | 2.9 ± 1.1 | 3.0 ± 1.1 | 2.9 ± 1.1 | 2.8 ± 1.1 | 2.8 ± 1.0 | 2.8 ± 1.1 |
| Triglycerides (mmol/L) | 1.4 (1.0–2.0) | 1.4 (1.0–2.0) | 1.4 (1.0–2.0) | 1.4 (1.0–2.0) | 1.4 (1.0–2.1) | 1.4 (1.0–2.0) |
| Estimated GFR (ml/min/1.73m2) | 76 ± 18 | 77 ± 19 | 76 ± 17 | 77 ± 17 | 76 ± 18 | 77 ± 19 |
| CRP (mg/L) | 2.1 (2.1–4.4) | 2.4 (1.1–4.9) | 2.0 (1.0–4.2) | 1.9 (0.9–4.0) | 1.9 (0.9–4.5) | 2.1 (1.0–4.5) |
| *Medication use* | | | | | | |
| Lipid lowering | 5091 (67%) | 981 (65%) | 994 (66%) | 1033 (68%) | 1039 (69%) | 1044 (69%) |
| Platelet inhibitor | 5762 (76%) | 1109 (73%) | 1165 (77%) | 1141 (75%) | 1184 (78%) | 1163 (77%) |
| Antihypertensives | 5599 (74%) | 1105 (73%) | 1061 (70%) | 1093 (72%) | 1164 (77%) | 1176 (78%) |
| Diuretics | 1574 (21%) | 467 (31%) | 305 (20%) | 251 (17%) | 262 (17%) | 289 (19%) |
| Loop diuretics | 617 (8%) | 253 (17%) | 109 (7%) | 82 (5%) | 89 (6%) | 84 (6%) |
| Thiazide diuretics | 874 (12%) | 191 (13%) | 178 (12%) | 159 (11%) | 156 (10%) | 190 (13%) |
| ACE-inhibitors | 2298 (30%) | 523 (35%) | 419 (28%) | 475 (31%) | 442 (29%) | 439 (29%) |
| Beta-blockers | 4023 (53%) | 751 (50%) | 738 (49%) | 838 (55%) | 863 (57%) | 833 (55%) |
| Calcium antagonists | 1568 (21%) | 278 (18%) | 268 (18%) | 268 (18%) | 323 (21%) | 431 (29%) |

All data in n (%) or mean ± standard deviation (except for triglycerides and CRP: median with IQR). HDL, high-density lipoprotein; LDL, low-density lipoprotein; Hs-CRP, high-sensitivity C-reactive protein; BMI, body mass index; eGFR, estimated glomerular filtration rate (calculated with Chronic Kidney Disease Epidemiology Collaboration [CKD-EPI] formula).

included linear terms for potassium urinary excretion. In the fully adjusted models, potassium urinary excretion was observed to have a positive relation with the primary composite outcome (MI, stroke, and cardiovascular mortality) (HR 1.25; 95%CI 1.13–1.39) (Fig 1C) and the separate components myocardial infarction (HR 1.26; 95%CI 1.07–1.48) and cardiovascular mortality (HR 1.20; 95%CI 1.06–1.37) (S2–S4 Figs). Also, potassium urinary excretion was positively associated with all-cause mortality (HR 1.13; 95%CI 1.03–1.25) (Fig 1D).

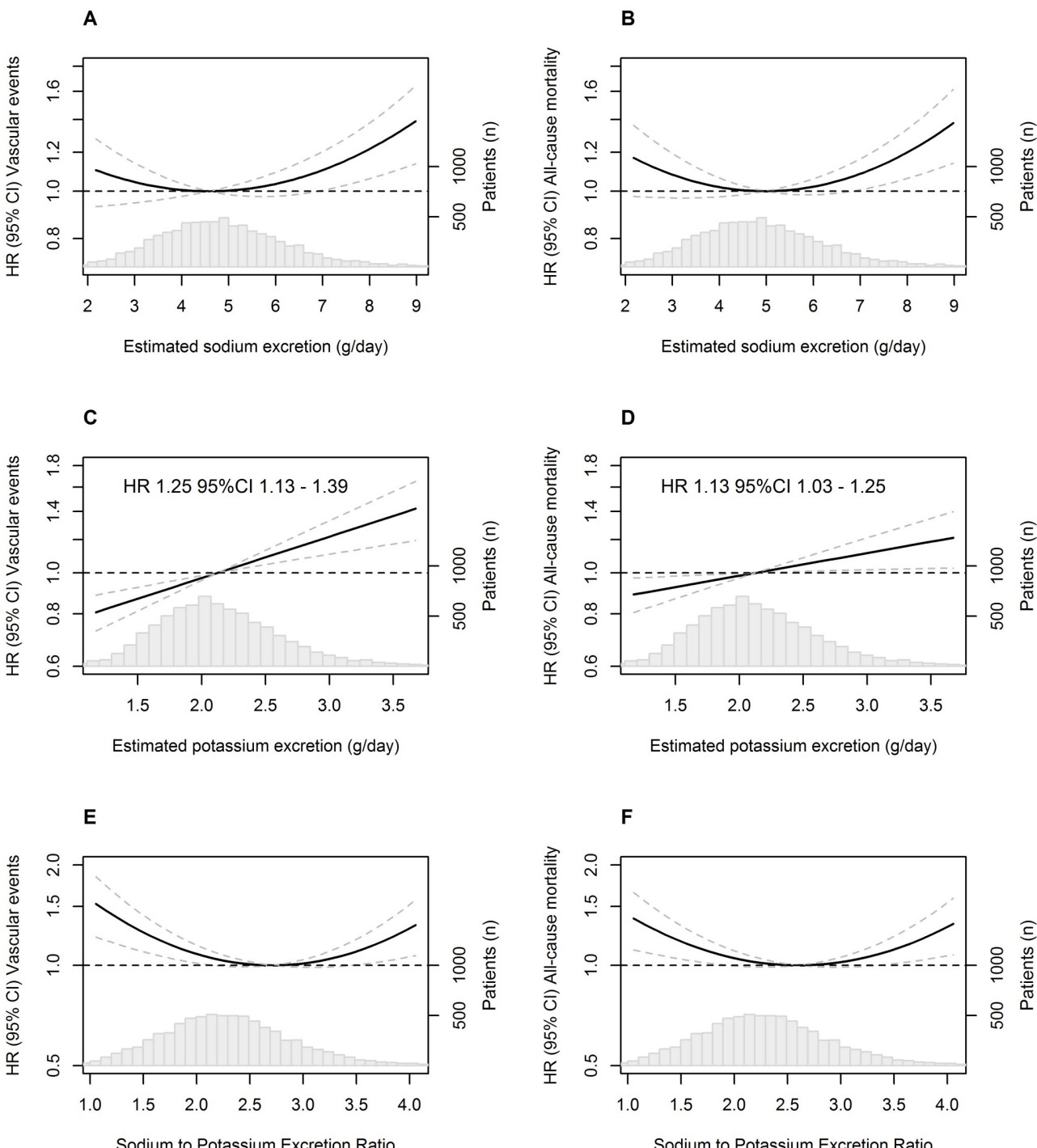

**Fig 1. Relation between salt excretion and recurrent cardiovascular events and mortality.** Adjusted hazard ratios for vascular events and mortality by baseline estimated salt excretion (distribution shown by histogram) A. Relation between estimated 24-hour urinary sodium excretion and vascular events (linear term P = 0.02; non-linear term P<0.01). Nadir: 4.59 g/day. B. Relation between estimated 24-hour urinary sodium excretion and mortality (linear term P<0.01; non-linear term <0.01). Nadir: 4.97 g/day. C. Relation between 1 gram/day higher estimated 24-hour urinary potassium excretion and vascular events. D. Relation between 1 gram/day higher estimated 24-hour urinary potassium excretion and mortality. E. Relation between sodium-to-potassium excretion ratio and vascular events (linear term P<0.01; non-linear term <0.01). Nadir: 2.71 g/day. F. Relation between sodium-to-potassium excretion ratio and mortality (linear term P<0.01; non-linear term <0.01). Nadir: 2.60 g/day. All hazard ratios were plotted between the 1st and 99th percentile of the corresponding salt measure. Dotted lines represent 95% confidence intervals. All models were adjusted for age, sex, current smoking, BMI (kg/m²), presence of diabetes, eGFR, and non-high-density lipoprotein cholesterol. HR = Hazard ratio.

### Relation between sodium-to-potassium excretion ratio and recurrent cardiovascular events and all-cause mortality

The relationship between sodium-to-potassium excretion ratio and the incidence of vascular events followed a J-shaped curve, with increased hazard rates at low and high ratios (Fig 1E; p<0.01; non-linear term p<0.01). Also, the relationship between sodium-to-potassium excretion ratio and all-cause mortality followed a J-shaped curve (Fig 1F; p<0.01; non-linear term p<0.01). The nadir for vascular events was 2.71 and 2.60 for all-cause mortality. No association was found between the sodium-to-potassium excretion ratio and the occurrence of stroke (p = 0.72, non-linear term p = 0.52) (S2 Fig) and the occurrence of myocardial infarction (p = 0.14; non-linear term p = 0.23) (S3 Fig). Still, the relationship between sodium-to-potassium excretion ratio and vascular mortality was J-shaped (p<0.01, non-linear term p<0.01, nadir 2.64) (S4 Fig).

### Interactions

Results of the interaction tests are shown in S4 Table. The effect of sodium-to-potassium excretion ratio on all-cause mortality was modified by age (<65 versus ≥65 years). Hence, results were stratified according to age (S5 Fig). In patients aged ≥65 years, the sodium-to-potassium excretion ratio was not associated with all-cause mortality. There were no other significant interaction terms.

### Sensitivity analysis

The shape of the relationship between sodium and potassium urinary excretion and vascular events and mortality did not materially change after exclusion of patients who experienced events or died within 1, 2, and 5 year(s) after inclusion and after exclusion of patients treated with loop diuretics (n = 617) (a surrogate for heart failure patients) (S6 and S7 Figs). In the first years of follow-up, survival rates for patients in the lower quintiles of salt excretion were similar to those of patients in the other quintiles of salt excretion (S8 Fig).

## Discussion

In the current study we found a J-shaped relation between estimated 24-hour sodium urinary excretion and recurrent vascular events and mortality in patients with vascular disease. The optimum estimated sodium urinary excretion found was between 4.5 grams per day and 5.0 grams per day, which is generally viewed as an excess in sodium intake. This J-shaped relation was even more pronounced when accounting for potassium intake, using the sodium-to-potassium excretion ratio, with an optimum ratio between 2.5 and 3.0. Increasing values of estimated 24-hour potassium urinary excretion increased the risk of recurrent vascular events and mortality, and this relation was linear.

Several previous observational studies in populations at high cardiovascular risk have also found a J-shaped curve between sodium urinary excretion levels and the risk of CVD and mortality [24–26]. In line with our findings, an observational post hoc analysis of 28,880 participants of the ONTARGET and TRANSCEND trials with established CVD or high-risk diabetes mellitus found a sodium excretion between 4 and 5.99 gram per day as the optimum level of sodium excretion using cardiovascular death, myocardial infarction, stroke, and hospitalization for congestive heart failure as outcome [24]. Studies in patients with diabetes (type 1 and 2) also found lower 24-hour urinary sodium excretion to be associated with increased cardiovascular [25] and all-cause mortality [25, 26]. Results from the current study add to the limited amount of evidence on the relation between sodium and cardiovascular events and mortality in a population with vascular disease.

Reverse causality has been proposed as an explanation for the relation observed between low sodium excretion and vascular events and mortality [27]. Observations suggestive of reverse causality include that a J-shaped association is seen during short, but not during prolonged follow-up [28] or that an initially present J-shaped relation becomes linear after exclusion of study participants having conditions that lead to reduced sodium intake and are simultaneously associated with an increased risk of adverse events. Sensitivity analyses of the present study showed that exclusion of patients with events within 1, 2, and 5 year(s) after start of the study and exclusion of patients treated with loop diuretics, considered as a proxy for a diagnosis of congestive heart failure, did not materially alter the shape of the relations. Still, we recognize that reverse causality cannot be completely ruled out and may partly account for the increased risk observed in patients with low sodium excretion.

Second, systematic error in sodium measurement has been proposed as an explanation for the paradoxical U- or J-shape relation [29]. Similar to this study, previous cohort studies often used formulas to estimate an individual's usual sodium intake based on a single spot urine rather than multiple non-consecutive 24-hour urine collections [30, 31]. Although the latter is cumbersome and logistically more challenging, the formula-based approach may result in systematic errors with overestimation at lower levels and underestimation at higher levels of sodium intake [32, 33]. This may even change the shape of the dose-response curve; placing subjects in poor health into groups with low sodium intake and falsely ascribe higher mortality to low sodium [33]. Although, a J-shaped relationship was also described in studies that measured sodium intake by 24-hour urine collections [9, 26], it can not be ruled out that the formula-based approach may in part lead to these paradoxical findings.

Third, it is also possible that the J-shaped relation is due to selection on the index event [34]. This can be understood by considering the onset of vascular events as the sum of the effect of multiple causal factors. If one important causal risk factor such as high sodium intake is already present, less effect of other factors is required for disease onset. Subsequently, comparing high sodium consumers with low sodium consumers who already have developed vascular disease, leads to the high sodium consumers having a relatively healthy risk profile compared to low sodium consumers in both measured and unmeasured factors. Nonetheless, the observed associations in this study remained after adjustment for most known risk factors for vascular disease, making index event bias a less likely explanation.

Besides methodological explanations, a causal mechanism explaining the relation observed between low sodium excretion and vascular events and mortality should also be considered. Sodium is an important electrolyte in the extracellular fluid and has an essential role in regulating the intra- and extracellular fluid. Previous neuroscience studies in animals have revealed neural networks that play a role in the regulation of sodium appetite to ensure a certain level of sodium intake [35]. From these studies, it is hypothesized that sodium is under strict control, which is supported by the observation that sodium is often within a narrow range. For example, the mean estimated 24-hour sodium excretion level in our study is close to the mean range for sodium intake defined by previous analyses of worldwide 24-hour urinary sodium excretion data [36–38]. Low sodium intake may therefore result in activation of a physiological mechanism to balance sodium concentration including an increase in plasma renin activity and aldosterone which consequently increase in sympathetic nerve activity [39], serum cholesterol and triglyceride levels, adrenalin secretion [40], and resistance to insulin [41, 42], which may counteract the benefit of lowering blood pressure.

In the current study, a positive linear relationship between estimated 24-hour urinary potassium excretion and the risk of recurrent MACE and all-cause mortality was observed. Considering the separate components of MACE, the effect of potassium excretion on recurrent MACE was mainly driven by an increased risk of myocardial infarction. These findings differ

compared to previous studies in primary and secondary prevention cohorts describing non-significant associations between potassium intake and coronary heart disease and significant inverse associations between potassium intake and MACE, respectively [13, 43, 44]. The discrepancies between our study and previous studies may be due to the difference in case-mix (patients with versus without vascular disease) and use of different statistical approaches. For example, previous studies were able to adjust for additional lifestyle factors (i.e. caloric, fruit, and vegetable intake), which reduced the risk of residual confounding [24]. However, these studies often analyzed 24-hour urinary potassium excretion categorically rather than continuously (using non-linear terms), potentially leading to a loss of power and inaccurate estimations [45, 46]. Moreover, reverse causality and index events bias may also have played a role here. However, sensitivity analyses evaluating the likelihood of these biases showed similar results, making these explanations less likely.

As with all studies of observational nature, no definitive causal conclusions can be drawn. To guide clinical practice, these findings need to be replicated by large and long-term randomized controlled trials evaluating the effect of different targets for dietary salt intake on clinical (cardiovascular) outcomes in patients with clinically manifest vascular disease. In the recently published Salt Substitute and Stroke Study (SSaSS) [47], involving 20.995 persons with either a history of stroke or a high BP from 600 villages in rural China, the effect of regular salt (100% sodium chloride) was compared with a salt substitute (75% sodium chloride and 25% potassium chloride) with respect to stroke. The combined use of lower sodium and higher potassium, by means of this substitute, led to a lower rate of stroke than the use of regular salt (rate ratio 0.86; 95%CI 0.77–0.96). Although SSaSS provides some answers, it remains unclear whether the effect can be attributed to lower sodium intake, higher potassium intake or both.

Strengths of the present cohort study include the large number of patients with manifest vascular disease with extensive and standardized measurement of risk factors at baseline and a long follow-up with a low proportion of patients lost to follow-up. Furthermore, the generalizability of the results is high as the UCC-SMART cohort consists of a referred patient population with a broad spectrum of vascular disease. A limitation of the study includes the possibility of measurement error when using the Kawasaki formulas for the conversion of spot urine sodium and potassium measurements into estimated 24-hour urinary excretion. Since a lower proportion (~77%) of ingested potassium is excreted renally [48], the estimated 24-hour urinary potassium excretion in this study is likely a suboptimal reflection of actual potassium intake in this population. Lastly, patient characteristics were only measured at baseline which made it unable to address the time-varying nature of sodium and potassium excretion.

In conclusion, in this observational study, relations between both estimated 24-hour sodium urinary excretion and sodium-to-potassium excretion ratio and recurrent MACE and all-cause mortality were J-shaped, with sodium excretion above and below 4.5–5.0 both being associated with higher risk of recurrent MACE and all-cause mortality. Furthermore, higher estimated 24-hour potassium urinary excretion was associated with a higher risk of recurrent MACE, mainly driven by an increased risk of myocardial infarction, and all-cause mortality. These results provide no evidence for dietary sodium restriction to levels between 1.5 and 2.4 g per day as a means of reducing the risk of recurrent CVD in patients with vascular disease and underline the need for further investigation into the relation between salt intake and cardiovascular disease in this population.

## Supporting information

**S1 Fig. Restricted-cubic-spline plots of the association between estimated salt excretion and recurrent major adverse cardiovascular events and all-cause mortality.** Restricted-

cubic-spline plots of association between estimated 24-hour urinary excretion of sodium (A-B), potassium (C-D), and their ratio (E-F) and recurrent MACE (left column) and all-cause mortality (right column). Histograms demonstrate distributions of different salt measures. The median of each salt measure (4.80 g/day, 2.12 g/day and 2.27 for sodium, potassium and their ratio, respectively) was taken as a reference (HR = 1.0). Spline curves were plotted between the 1st and 99th percentile of the corresponding salt measure. Dotted lines represent 95% confidence intervals. All plots were adjusted for age, sex, current smoking, BMI (kg/m2), presence of diabetes, eGFR, and non-high-density lipoprotein cholesterol. HR = Hazard ratio. (PNG)

**S2 Fig. Relationship between salt excretion and the occurrence of stroke.** A. Relation between 1 gram/day higher estimated 24-hour urinary sodium excretion and the occurrence of stroke. B. Relation between 1 gram/day higher estimated 24-hour urinary potassium excretion and the occurrence of stroke. C. Relation between 1 unit higher sodium-to-potassium excretion ratio and the occurrence of stroke. Histograms demonstrate distributions of different salt measures. The median of each salt measure (4.80 g/day, 2.12 g/day and 2.27 for sodium, potassium and their ratio, respectively) was taken as a reference (HR = 1.0). All hazard ratios were plotted between the 1st and 99th percentile of the corresponding salt measure. Dotted lines represent 95% confidence intervals. All plots were adjusted for age, sex, current smoking, BMI (kg/m2), presence of diabetes, eGFR, and non-high-density lipoprotein cholesterol. HR = Hazard ratio. (PNG)

**S3 Fig. Relationship between salt excretion and the occurrence of myocardial infarction.** A. Relation between 1 gram/day higher estimated 24-hour urinary sodium excretion and the occurrence of myocardial infarction. B. Relation between 1 gram/day higher estimated 24-hour urinary potassium excretion and the occurrence of myocardial infarction. C. Relation between 1 unit higher sodium-to-potassium excretion ratio and the occurrence of myocardial infarction. Histograms demonstrate distributions of different salt measures. The median of each salt measure (4.80 g/day, 2.12 g/day and 2.27 for sodium, potassium and their ratio, respectively) was taken as a reference (HR = 1.0). All hazard ratios were plotted between the 1st and 99th percentile of the corresponding salt measure. Dotted lines represent 95% confidence intervals. All plots were adjusted for age, sex, current smoking, BMI (kg/m2), presence of diabetes, eGFR, and non-high-density lipoprotein cholesterol. HR = Hazard ratio. (PNG)

**S4 Fig. Relationship between salt excretion and vascular mortality.** A. Relation between estimated 24-hour urinary sodium excretion and vascular mortality (linear term P<0.01; non-linear term P<0.01). Nadir: 4.98 g/day. B. Relation between 1 gram/day higher estimated 24-hour urinary potassium excretion and vascular mortality. C. Relation between sodium-to-potassium excretion ratio and vascular mortality (linear term P<0.01, non-linear term P<0.01). Nadir 2.64. Histograms demonstrate distributions of different salt measures. All hazard ratios were plotted between the 1st and 99th percentile of the corresponding salt measure. Dotted lines represent 95% confidence intervals. All plots were adjusted for age, sex, current smoking, BMI (kg/m2), presence of diabetes, eGFR, and non-high-density lipoprotein cholesterol. HR = Hazard ratio. (PNG)

**S5 Fig. Stratified analyses for patients <65 years and ≥65 years of age.** Adjusted hazard ratio for mortality by baseline sodium-to-potassium excretion ratio. Hazard ratios were plotted between the 1st and 99th percentile of the sodium-to-potassium excretion ratio. Plots were

adjusted for age, sex, current smoking, BMI (kg/m2), presence of diabetes, eGFR, and non-high-density lipoprotein cholesterol. HR = Hazard ratio.
(PNG)

**S6 Fig. Sensitivity analysis excluding patients with short follow-up.** A-B. Change in estimated effect between estimated 24-hour sodium urinary excretion and vascular events (A) and mortality (B) after exclusion of patients who experienced events or died within 1 year (dashed red line), 2 years (dashed green line), and 5 years (dashed blue line) after inclusion. Black lines depict the main analysis. C-D. Change in estimated effect between 24-hour potassium urinary excretion and vascular events (C) and mortality (D) after exclusion of patients who experienced events or died within 1 year (dashed red line), 2 years (dashed green line), and 5 years (dashed blue line) after inclusion. E-F. Change in estimated effect between sodium-to-potassium excretion ratio and vascular events (E) and mortality (F) after exclusion of patients who experienced events or died within 1 year (dashed red line), 2 years (dashed green line), and 5 years (dashed blue line) after inclusion. HR = Hazard ratio.
(PNG)

**S7 Fig. Sensitivity analysis excluding patients treated with loop diuretics.** A-B. Change in estimated effect between estimated 24-hour sodium urinary excretion and vascular events (A) and mortality (B) after exclusion of patients who were treated with loop diuretics (dashed blue line). Black lines depict the main analysis. C-D. Change in estimated effect between 24-hour potassium urinary excretion and vascular events (C) and mortality (D) after exclusion of patients who were treated with loop diuretics (dashed blue line). E-F. Change in estimated effect between sodium-to-potassium excretion ratio and vascular events (E) and mortality (F) after exclusion of patients who were treated with loop diuretics (dashed blue line). HR = Hazard ratio.
(PNG)

**S8 Fig. Sensitivity analysis evaluating survival curves for quintiles of salt excretion.** A-B. Survival curves in quintiles of estimated 24-hour sodium excretion for (A) recurrent cardiovascular disease; (B) all-cause mortality. C-D. Survival curves in quintiles of estimated 24-hour potassium excretion for (C) recurrent cardiovascular disease; (D) all-cause mortality. E-F. Survival curves in quintiles of the sodium-to-potassium ratio for (E) recurrent cardiovascular disease; (F) all-cause mortality.
(PNG)

**S1 Table. Kawasaki formula used to predict 24-hour urinary sodium and potassium excretion from spot urine samples.**
(DOCX)

**S2 Table. Definitions of vascular outcomes.**
(DOCX)

**S3 Table. Baseline characteristics of all participants, according to estimated 24 hour potassium excretion.**
(DOCX)

**S4 Table. P-values for interaction.**
(DOCX)

## Acknowledgments

We gratefully acknowledge the contribution of the research nurses; R. van Petersen (data-manager); B. van Dinther (study manager), and the members of the Utrecht Cardiovascular

Cohort-Second Manifestations of ARTerial disease-Studygroup (UCC-SMART-Studygroup): F.W. Asselbergs and H.M. Nathoe, Department of Cardiology; G.J. de Borst, Department of Vascular Surgery; M.L. Bots and M.I. Geerlings, Julius Center for health Sciences and Primary Care; M.H. Emmelot, Department of Geriatrics; P.A. de Jong and T. Leiner, Department of Radiology; A.T. Lely, Department of Obstetrics & Gynecology; N.P. van der Kaaij, Department of Cardiothoracic Surgery; L.J. Kappelle and Y.M. Ruigrok, Department of Neurology; M.C. Verhaar, Department of Nephrology, F.L.J. Visseren (**chair**; email address: F.L.J.Visseren-n@umcutrecht.nl) and J. Westerink, Department of Vascular Medicine, University Medical Center Utrecht and Utrecht University.

## Author Contributions

**Conceptualization:** Eline H. Groenland, Jean-Paul Vendeville, Michiel L. Bots, Frank L. J. Visseren, Wilko Spiering.

**Data curation:** Eline H. Groenland, Jean-Paul Vendeville.

**Formal analysis:** Eline H. Groenland.

**Investigation:** Eline H. Groenland, Jean-Paul Vendeville.

**Methodology:** Eline H. Groenland, Jean-Paul Vendeville, Michiel L. Bots, Frank L. J. Visseren, Wilko Spiering.

**Project administration:** Michiel L. Bots, Frank L. J. Visseren, Wilko Spiering.

**Supervision:** Michiel L. Bots, Frank L. J. Visseren, Wilko Spiering.

**Visualization:** Eline H. Groenland.

**Writing – original draft:** Eline H. Groenland.

**Writing – review & editing:** Jean-Paul Vendeville, Michiel L. Bots, Gert Jan de Borst, Hendrik M. Nathoe, Ynte M. Ruigrok, Peter J. Blankestijn, Frank L. J. Visseren, Wilko Spiering.

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
