## [Decision Letter · Decision Letter 0]

10 Jan 2022

PONE-D-21-34821The relation between urinary sodium and potassium excretion and risk of cardiovascular events and mortality in patients with cardiovascular diseasePLOS ONE

Dear Dr. Spiering,

Thank you for submitting your manuscript to PLOS ONE. After careful consideration, we feel that it has merit but does not fully meet PLOS ONE’s publication criteria as it currently stands. Therefore, we invite you to submit a revised version of the manuscript that addresses the points raised during the review process.

As you will recognize from the comments of the reviewer major points of critique were raised, especially regarding study design and conclusions drawn from your findings.

Please submit your revised manuscript within Feb 24 2022 11:59PM. If you will need more time than this to complete your revisions, please reply to this message or contact the journal office at plosone@plos.org. Please include the following items when submitting your revised manuscript:A rebuttal letter that responds to each point raised by the academic editor and reviewer(s). You should upload this letter as a separate file labeled 'Response to Reviewers'.A marked-up copy of your manuscript that highlights changes made to the original version. You should upload this as a separate file labeled 'Revised Manuscript with Track Changes'.An unmarked version of your revised paper without tracked changes. You should upload this as a separate file labeled 'Manuscript'.

We look forward to receiving your revised manuscript.

Kind regards,

Rudolf Kirchmair

Academic Editor

PLOS ONE

2. One of the noted authors is a group or consortium “UCC-SMART study group”. In addition to naming the author group, please list the individual authors and affiliations within this group in the acknowledgments section of your manuscript. Please also indicate clearly a lead author for this group along with a contact email address.

- https://www.ajconline.org/article/S0002-9149(16)31119-5/fulltext

- https://link.springer.com/article/10.1007%2Fs40620-021-00996-1

The text that needs to be addressed involves the Results.

In your revision ensure you cite all your sources (including your own works), and quote or rephrase any duplicated text outside the methods section. Further consideration is dependent on these concerns being addressed.

Reviewers' comments:

Reviewer's Responses to Questions

**Comments to the Author**

1. Is the manuscript technically sound, and do the data support the conclusions?

Reviewer #1: Yes

2. Has the statistical analysis been performed appropriately and rigorously? 

Reviewer #1: Yes

3. Have the authors made all data underlying the findings in their manuscript fully available?

Reviewer #1: Yes

4. Is the manuscript presented in an intelligible fashion and written in standard English?

Reviewer #1: Yes

5. Review Comments to the Author

Reviewer #1: The sample size of the study is quite large. The manuscript is generally well written. There are several suggestions for revision of the manuscript.

1. When mortality and to some extent also cardiovascular events are considered in a study on dietary intakes of sodium or other nutrients, there is always possibility of reverse causality. The lower side of the J-curve can be consequence of reverse causality. There are two approached to deal with this issue. First, the authors may do a survival analysis and show the survival curve to see whether there is reverse causality in patients with low sodium excretion. Second, the authors may look at the hazard of cardiovascular events and all-cause mortality at least one year after entry into the study. Those with worse health and low sodium intake may likely develop a non-fatal or fatal event within a year of entry.

2. The authors studied the estimated excretion of sodium and potassium and the sodium to potassium ratio. These measurements are highly related but have very different clinical or pathogenic implications. Sodium excretion is largely a measure of its intake, but potassium excretion is less so. The authors may need to deal with this very carefully.

3. There are several formulae for the estimation of sodium excretion on the basis of spot urine sodium concentration. The authors may need to justify their choice of the Kawasaki formula but not others.

4. The manuscript is long, and may need to shorten some parts, such as "Discussion".

5. The figures have a low resolution, and need to be reproduced.

6. PLOS authors have the option to publish the peer review history of their article (what does this mean?). If published, this will include your full peer review and any attached files.

Reviewer #1: **Yes: **Ji-Guang Wang

---

## [Author Response · Author response to Decision Letter 0]

27 Jan 2022

Letter Revised Manuscript

Date January 2022

Regarding Submission revision

Title The relation between urinary sodium and potassium excretion and risk of cardiovascular events and mortality in patients with cardiovascular disease

ID PONE-D-21-34821

First of all, we would like to thank the reviewer for the thorough review of the manuscript. The issues raised are all very valuable and we are very pleased with the improvements that have resulted from these comments. Below is a detailed response to the comments of the reviewers. Page numbers refer to the article file “Revised Manuscript with Track Changes”

Reviewers' comments:

Review Comments to the Author

The sample size of the study is quite large. The manuscript is generally well written. 

We would like to thank the reviewer for the positive feedback.

There are several suggestions for revision of the manuscript.

1. When mortality and to some extent also cardiovascular events are considered in a study on dietary intakes of sodium or other nutrients, there is always possibility of reverse causality. The lower side of the J-curve can be consequence of reverse causality. There are two approached to deal with this issue. First, the authors may do a survival analysis and show the survival curve to see whether there is reverse causality in patients with low sodium excretion. Second, the authors may look at the hazard of cardiovascular events and all-cause mortality at least one year after entry into the study. Those with worse health and low sodium intake may likely develop a non-fatal or fatal event within a year of entry.

We recognize that reverse causality is among the most important biases in observational studies. Therefore, we already performed a sensitivity analysis in which we excluded patients who experienced recurrent cardiovascular disease (CVD) within 1, 2, and 5 year(s) after inclusion to eventually evaluate the hazard of recurrent CVD and all-cause mortality at least one year after entry into the study. For the results of these analyses we would like to refer to Supplemental Figure S6 in the supporting information. Compared to the main analyses, the results did not differ substantially, suggesting that the risk of reverse causality is low. 

We would like to thank the reviewer for the suggestion to perform a survival analysis in patients with low sodium excretion. For this analysis, we first stratified patients by quintiles of each salt measure (estimated 24-hour sodium excretion, estimated potassium excretion, and sodium-to-potassium ratio). Next, Kaplan-Meier survival curves were plotted per quintile of each salt measure for recurrent CVD and all-cause mortality. Results are displayed in the figure below. 

Based on this figure, it can be concluded that in the first years of follow-up, the survival rate of patients in the lowest quintiles of each salt measure is similar to the other quintiles. This suggests that the risk of reverse causality is small.

We included this figure in the supporting information, page 13, as supplemental figure 8. Moreover, we added to the method section, page 8, line 177-180: “Lastly, to evaluate whether patients with low levels of salt excretion had lower survival rates in the first years of follow-up, Kaplan-Meier survival curves were plotted by quintile of each salt measure (estimated 24-hour sodium excretion, estimated potassium excretion, and stage-to-potassium ratio) for recurrent CVD and all-cause mortality.” and to the results section, page 13, line 309-311: “In the first years of follow-up, survival rates for patients in the lower quintiles of salt excretion were similar to those of patients in the other quintiles of salt excretion (Supplemental Figure S8).”

2. The authors studied the estimated excretion of sodium and potassium and the sodium to potassium ratio. These measurements are highly related but have very different clinical or pathogenic implications. Sodium excretion is largely a measure of its intake, but potassium excretion is less so. The authors may need to deal with this very carefully.

We agree with the reviewer that urine sodium excretion is a better measure of actual sodium intake than urine potassium excretion is for actual potassium intake. Under homeostatic circumstances of constant sodium intake, approximately 93% of ingested sodium is excreted in the urine, whereas for potassium this only accounts for 77% (1,2). We addressed this issue by including it as a limitation in the discussion section, page 18, line 429-431: 

“ Since a lower proportion (~77%) of ingested potassium is excreted renally (49), the estimated 24-hour urinary potassium excretion in this study is likely a suboptimal reflection of actual potassium intake in this population.”

3. There are several formulae for the estimation of sodium excretion on the basis of spot urine sodium concentration. The authors may need to justify their choice of the Kawasaki formula but not others.

We thank the reviewer for pointing this out. Indeed, there are several other formulas available to estimate 24-hour salt excretion. We used the Kawasaki formula to estimate 24-hour urinary sodium and potassium excretion because it has previously been shown to provide the most valid estimate of sodium intake in different populations (in comparison to other formula-based approaches). In addition, it is the most commonly used formula in previous studies that also evaluated the relationship between estimated salt excretion and CVD and/or all-cause mortality. The use of the Kawasaki formula thus allowed us to compare our results with previous research. 

We added the following sentence to the method section, page 5, line 144-147: 

“We chose to use the Kawasaki formula to allow comparability between this and previous studies and because this formula is considered the least biased method for estimating 24-hour sodium excretion compared to other formula-based approaches (3).” 

4. The manuscript is long, and may need to shorten some parts, such as "Discussion".

We have thoroughly read through the entire manuscript and checked it for redundant parts. We aimed to present and discuss the findings as concisely as possible. 

5. The figures have a low resolution, and need to be reproduced.

At submission, we uploaded the figures in high resolution (600 dpi, according to PLOS ONE’s style requirements). However, in order to download the entire submission file as quickly as possible, the compiled PDF file includes low-resolution preview images of the figures after the reference list. To download a high-resolution version of each figure I would like to recommend the reviewer to click on the link at the top of each preview page.

 

Recommendations of the PLOS ONE’s Style Guide were followed and adjusted in the revised manuscript appropriately.

2. One of the noted authors is a group or consortium “UCC-SMART study group”. In addition to naming the author group, please list the individual authors and affiliations within this group in the acknowledgments section of your manuscript. Please also indicate clearly a lead author for this group along with a contact email address.

We have included a list of individual authors belonging to the UCC-SMART study group in the Acknowledgements section. The lead author of this group is prof. dr. F.L.J. Visseren (email address: F.L.J.Visseren@umcutrecht.nl), which is also indicated in the Acknowledgement section. 

- https://www.ajconline.org/article/S0002-9149(16)31119-5/fulltext

- https://link.springer.com/article/10.1007%2Fs40620-021-00996-1

The text that needs to be addressed involves the Results.

In your revision ensure you cite all your sources (including your own works), and quote or rephrase any duplicated text outside the methods section. Further consideration is dependent on these concerns being addressed.

The studies referenced are those also conducted in the UCC-SMART population. It could therefore be that certain parts of the method have similarities with these earlier studies. We have compared the manuscript with these two earlier manuscripts and found that there was no exact overlap.

 

References 

1. Lucko AM, Doktorchik C, Woodward M, Cogswell M, Neal B, Rabi D, et al. Percentage of ingested sodium excreted in 24-hour urine collections: A systematic review and meta-analysis. J Clin Hypertens (Greenwich). 2018 Sep;20(9):1220–9. 

2. Holbrook JT, Patterson KY, Bodner JE, Douglas LW, Veillon C, Kelsay JL, et al. Sodium and potassium intake and balance in adults consuming self-selected diets. Am J Clin Nutr. 1984 Oct;40(4):786–93. 

3. Mente A, O’Donnell MJ, Dagenais G, Wielgosz A, Lear SA, McQueen MJ, et al. Validation and comparison of three formulae to estimate sodium and potassium excretion from a single morning fasting urine compared to 24-h measures in 11 countries. J Hypertens. 2014 May;32(5):1005–14; discussion 1015.

---

## [Decision Letter · Decision Letter 1]

2 Mar 2022

The relation between urinary sodium and potassium excretion and risk of cardiovascular events and mortality in patients with cardiovascular disease

PONE-D-21-34821R1

Dear Dr. Spiering,

We’re pleased to inform you that your manuscript has been judged scientifically suitable for publication and will be formally accepted for publication once it meets all outstanding technical requirements.

Kind regards,

Rudolf Kirchmair

Academic Editor

PLOS ONE

Additional Editor Comments (optional):

Reviewers' comments:

Reviewer's Responses to Questions

**Comments to the Author**

1. If the authors have adequately addressed your comments raised in a previous round of review and you feel that this manuscript is now acceptable for publication, you may indicate that here to bypass the “Comments to the Author” section, enter your conflict of interest statement in the “Confidential to Editor” section, and submit your "Accept" recommendation.

Reviewer #1: All comments have been addressed

2. Is the manuscript technically sound, and do the data support the conclusions?

Reviewer #1: Yes

3. Has the statistical analysis been performed appropriately and rigorously? 

Reviewer #1: Yes

4. Have the authors made all data underlying the findings in their manuscript fully available?

Reviewer #1: Yes

5. Is the manuscript presented in an intelligible fashion and written in standard English?

Reviewer #1: Yes

6. Review Comments to the Author

Reviewer #1: The authors adequately addressed all the comments of the Reviewer, and revised the manuscript properly. No further comment.

7. PLOS authors have the option to publish the peer review history of their article (what does this mean?). If published, this will include your full peer review and any attached files.

Reviewer #1: **Yes: **Ji-Guang Wang

---

## [Editor Report · Acceptance letter]

8 Mar 2022

PONE-D-21-34821R1 

The relation between urinary sodium and potassium excretion and risk of cardiovascular events and mortality in patients with cardiovascular disease 

Dear Dr. Spiering:

I'm pleased to inform you that your manuscript has been deemed suitable for publication in PLOS ONE. Congratulations! Your manuscript is now with our production department. 

Kind regards, 

on behalf of

Prof Rudolf Kirchmair 

Academic Editor

PLOS ONE